# First Report of *Anopheles annularis s.l.*, *An. maculatus s.s.*, and *An. culicifacies s.l.* as Malaria Vectors and a New Occurrence Record for *An. pseudowillmori* and *An. sawadwongporni* in Alipurduar District Villages, West Bengal, India

**DOI:** 10.3390/microorganisms12010095

**Published:** 2024-01-03

**Authors:** Jadab Rajkonwar, Varun Shende, Ananta Kumar Maji, Apoorva Pandey, Puran K. Sharma, Kasinathan Gunasekaran, Sarala K. Subbarao, Dibya Ranjan Bhattacharyya, Kamaraju Raghavendra, Rocky Pebam, Vijay Mayakrishnan, Phiroz Gogoi, Susmita Senapati, Pallabi Sarkar, Saurav Biswas, Daniel Debbarma, Tulika Nirmolia, Sasmita Rani Jena, Bahniman Bayan, Pinki Talukder, Ashwarya Kumari Sihag, Himadri Sankar Bharali, Anisha Verma, Kongkon Mahanta, Gonsalo Sumer, Ranjan Karmakar, Saurav Jyoti Patgiri, Supriya Chaudhuri, Sumit Ganguli, Harpreet Kaur, Tapas K. Bhattacharyya, Pyare Laal Joshi, Bidhan Goswami, Kalpana Baruah, Sanghamitra Pati, Kanwar Narain, Ipsita Pal Bhowmick

**Affiliations:** 1Regional Medical Research Centre, Northeast Region (RMRC-NE)-ICMR, Dibrugarh 786001, India; jjrajkonwar22@gmail.com (J.R.); shendevarun@gmail.com (V.S.); drbhattacharyya@yahoo.com (D.R.B.); mayakrishnanvijay@gmail.com (V.M.); phirozgogoi@gmail.com (P.G.); senapatitiku999@gmail.com (S.S.); spallabi85@gmail.com (P.S.); saurav.icmrrmrcne@gmail.com (S.B.); bahnimanbayan5@gmail.com (B.B.); tinatalukdar34@gmail.com (P.T.); ash6856@gmail.com (A.K.S.); himadrisbharali@gmail.com (H.S.B.); anisha.verma257@gmail.com (A.V.); kongkonmahanta@gmail.com (K.M.); gonsalo.icmr@gmail.com (G.S.); ranjankarmakar1996@gmail.com (R.K.); sjpatgiri.rmrcne@icmr.gov.in (S.J.P.); sanghamitra.pati@icmr.gov.in (S.P.); knarain.rmrcne@gov.in (K.N.); 2District Health & Family Welfare Samiti, Alipurduar 736121, India; anantamaji08@gmail.com (A.K.M.); dycmoh2.apd@gmail.com (S.C.); cmohapd@gmail.com (S.G.); 3Indian Council of Medical Research (ICMR), Ramalingaswami Bhavan, New Delhi 110029, India; apoorva.icmr@gmail.com (A.P.); kaurh.hq@icmr.gov.in (H.K.); 4Department of Health & Family Welfare, Govt of West Bengal, Alipurduar 736121, India; puran.sharma611@gmail.com; 5ICMR-Vector Control Research Centre, Indira Nagar, Puducherry 605006, India; k_guna@yahoo.com; 6ICMR-National Institute of Malaria Research, Sector-8, Dwarka, New Delhi 110077, India; kamarajur2000@yahoo.com; 7NorthEast Space Application Centre (NESAC), Department of Space, Government of India, Umiam 793103, India; rocky.pebam@gmail.com (R.P.); danieldebbarma46@gmail.com (D.D.); 8National Health Mission, Bishwanath Charali 784176, India; tulikanirmolia@gmail.com; 9Regional Office of Health and Family Welfare, Kolkata 700106, India; sjena4@gmail.com (S.R.J.); dr.tkbhatta@gmail.com (T.K.B.); 10Directorate of National Vector Borne Disease Control Programme, Ministry of Health and Family Welfare, Government of India, Delhi 110054, India; doctorjoshi00@gmail.com (P.L.J.); drkalpanabaruah@gmail.com (K.B.); 11Agartala Govt Medical College, Agartala 799006, India; mruagmc@gmail.com

**Keywords:** *An. annularis s.l.*, *An. maculatus s.s.*, *An. culicifacies*, *An. pseudowillmori*, *An. sawadwongporni*, first report, Alipurduar, India, malaria

## Abstract

A comprehensive entomological survey was undertaken in Alipurduar District, West Bengal, from 2018 to 2020 and in 2022. This study was prompted by reported malaria cases and conducted across nine villages, seven Sub-Centres, and three Primary Health Centres (PHCs). Mosquitoes were hand-collected with aspirators and flashlights from human dwellings and cattle sheds during the daytime. Both morphological and molecular techniques were used for species identification. Additionally, mosquitoes were tested for *Plasmodium* parasites and human blood presence. Mosquito species such as *An. barbirostris s.l.*, *An. hyrcanus s.l.*, *An. splendidus*, and *An. vagus* were morphologically identified. For species like *An. annularis s.l.*, *An. minimus s.s.*, *An. culicifacies s.l.*, and *An. maculatus s.s*., a combination of morphological and molecular techniques was essential. The mitochondrial cytochrome c oxidase gene subunit 1 (CO1) was sequenced for *An. annularis s.l.*, *An. maculatus s.s.*, *An. culicifacies s.l.*, *An. vagus*, and some damaged samples, revealing the presence of *An. pseudowillmori* and *An. fluviatilis*. The major *Anopheles* species were *An. annularis s.l*., *An. culicifacies s.l.*, and *An. maculatus s.s*., especially in Kumargram and Turturi PHCs. *Plasmodium* positivity was notably high in *An. annularis s.l.* and *An. maculatus s.s.* with significant human blood meal positivity across most species. Morphological, molecular, and phylogenetic analyses are crucial, especially for archived samples, to accurately identify the mosquito fauna of a region. Notably, this study confirms the first occurrence of *An. pseudowillmori* and *An. sawadwongporni* in West Bengal and implicates *An. maculatus s.s*., *An. culicifacies s.l.*, and *An. annularis s.l.* as significant vectors in the Alipurduar region.

## 1. Introduction

Alipurduar District, situated in West Bengal, India, and bordering Bhutan, forms part of the “chicken neck” corridor that links the Northeastern (NE) part of India with the rest of the country [1]. The importance of this district from the perspective of malaria incidence and vector pattern lies in its strategic location, both nationally and internationally. This district shares an extensive boundary with Bhutan, a nation actively pursuing malaria elimination, but there are only a few studies from the area and almost none in the past decade.

While the Annual Parasite Incidence (API) for the district, according to the National Centre for Vector Borne Disease Control (NCVBDC) data from 2017 to 2022, was consistently <1, certain areas like Kumargram and Turturi Primary Health Centres (PHCs) observed sporadic malaria outbreaks. Notably, a significant outbreak was reported in 2019 from the Raydak TE-1 Sub-Centre area under the Turturi PHC [1,2], emphasising the important details about vectors from these pockets (Data from District Health Department, also tabulated as Appendix A).

There are no studies on the cases that might have migrated from Bhutan to this part of India. There are earlier reports on the vectorial status of this region. For instance, *Anopheles minimus*, a major malaria vector in the NE region, was once widespread across the Himalayan foothills, only to disappear post-DDT treatments and subsequently resurface in the Dooars (Jalpaiguri and Alipurduar Districts of West Bengal, falling in the Himalayan region) area after many years [3]. Hence, there is a need for generating data that will help in understanding the malaria situation in the area and in planning the vector control strategy towards the proposed elimination. 

Earlier, *An. dirus* complex mosquitoes were reported from Jalpaiguri Dooars of West Bengal [4]. However, the exact species (sibling species) present in the *An. dirus* complex, *An. culicifacies* complex, *An. maculatus* complex, *An. minimus* complex, and *An. fluviatilis* complex is still unknown. 

A study spanning 2006-07 in the sub-Himalayan Dooars region of West Bengal, reportedly endemic for malaria, recorded anophelines such as *An. minimus*, *An. varuna*, *An. maculatus*, *An. fluviatilis*, *An. vagus*, *An. culicifacies*, *An. hyrcanus*, and *An. barbirostris* from huts and tea garden dwellings [5]. The vector status was however not reported in that study.

The heterogeneity in vector patterns, coupled with the region’s significance in connecting diverse malaria-prone zones, positions Alipurduar as a potential entomological hotspot. Past studies in West Bengal have identified vectors such as *An. philippinensis*, *An. varuna*, and *An. annularis s.l.*, but comprehensive data from the last 30 years, specifically from Alipurduar, remains scanty [6,7,8]. Moreover, modern molecular studies on vector status using the human blood index (HBI) to understand the importance of the species as a vector and precise *Anopheles* species identification are missing.

In light of these gaps, our pilot study aims to provide contemporary entomological insights from Alipurduar, focusing on vector species composition, the HBI, and the molecular identification of malaria parasites.

## 2. Materials and Methods

### 2.1. Study Location

Our research was conducted on mosquitoes that were randomly hand-collected from human dwellings and cattle sheds between 2018 and 2020 and in 2022. These collections took place across nine villages under the jurisdiction of seven Sub-Centres (namely, Dhowlabasti, Gopalpur T.G., Joydebpur, Raydak TE-1, Turturikhanda, Uttar Shibkata, and West Chengmari), spanning three Primary Health Centres (PHCs): Kumargram, Turturi, and Madhya Rangalibazna in Alipurduar District, West Bengal.

A land use and land cover (LULC) map of the study site was prepared. The vector data on the ward boundary for the study area was downloaded from the Survey of India portal (www.surveyofindia.gov.in accessed on 9 October 2023). The LULC map for the district was extracted from the Bhuvan web map services (www.bhuvan.nrsc.gov.in accessed on 9 October 2023), and the layer was updated using a standard false colour composite of the Sentinel-2 dataset pertaining to 2022 with a mapping scale of 1:25,000. LULC classes were prepared as per the standards of ISRO’s SIS-DP project (Space Based Information Support for Decentralised Planning), and the processing and composition of the map were made using ArcMap 10.8.1.

### 2.2. Collection of Mosquito Samples

All the mosquito samples were hand-collected with aspirators and flashlights from human dwellings and cattle sheds during the daytime. Each structure was searched for 15 min.

### 2.3. Identification of Mosquito Species

To identify the *Anopheles* species, we combined morphological and molecular methods. For species, namely, *An. annularis s.l.*, *An. minimus s.s*., *An. varuna*, *An. aconitus*, *An. maculatus s.s.*, and *An. sawadwongporni*, molecular methods were used alongside morphological methods. This was crucial because certain species within the same group or series are almost indistinguishable based on their physical characteristics alone.

### 2.4. Identification Based on Morphology

Mosquito species from the 2018–2020 and 2022 surveys were morphologically identified using the guidelines of Rattanarithikul and Green (1986) [9] for the *An. maculatus* group and Nagpal and Sharma (1995) [10] for all anopheline species.

### 2.5. Mosquito Dissection for Molecular Studies

Each mosquito was cleanly dissected into the head–thorax and abdomen to detect parasites and human blood, respectively. Additionally, a middle leg was detached for species identification with PCR. We extracted genomic DNA utilising the QiAmp DNA Extraction Kit (Qiagen, Germantown, CA, USA), adhering to the manufacturer’s guidelines.

### 2.6. Polymerase Chain Reaction Assays for Identifying Species

PCR reactions were performed in 20 μL volumes using a 2× Promega master mix (Promega, Madison, WI, USA) and approximately 50 ng of DNA template, executed on a Bio-rad C1000 thermal cycler (Hercules, CA, USA). We followed reference protocols for the thermal profile and primer concentration. Morphologically identified specimens underwent molecular analysis for validation, especially since many samples were old and lacked all identifying characteristics. The Walton et al. (2007) allele-specific PCR method was used to confirm Maculatus sibling species [11]. To identify mosquito species from the Funestus Group of the Myzomyia Series with very similar morphologies, Phuc et al. (2003) was used [12].

To differentiate species within the Annularis Group like *An. annularis*, *An. philippinensis*, and *An. nivipes*, the ribosomal DNA gene was targeted. A multiplex PCR approach was implemented, using a universal forward primer and four species-specific reverse primers [13]. The details of PCR primers used for all the molecular assays are given in Appendix A.

### 2.7. CO1 Sequencing for Identifying the Species

The mitochondrial Cytochrome Oxidase 1 (CO1) gene [14] was amplified and sequenced using an Applied Biosystem 3500 Genetic Analyzer for selected species deemed as vectors and damaged mosquitoes with incomplete characteristics. CO1 sequences underwent NCBI BLAST searches, with similar sequences sourced from the GenBank database. A maximum likelihood phylogenetic tree was prepared with MEGA X version 10.2.6., supported by a 500-bootstrap value. GenBank accession numbers OR620082-OR620089 (from this study) were included in the phylogenetic tree.

### 2.8. Human Blood Meal Detection

Regardless of their visible blood-fed status, all mosquitoes were tested for human blood presence using PCR. The method from Mohanty et al. (2007) was used by taking DNA template extracted from the abdomen [15].

### 2.9. Detection of Plasmodium in Mosquitoes

DNA was extracted from the head–thorax of individual *Anopheles* specimens to identify parasite sporozoites in mosquitoes. DNA extracted from the head–thorax of *Aedes* and *Culex* was used as a negative control. The nested quantitative PCR (nqPCR) for the cytochrome b gene was used as it was more sensitive than conventional PCR and could amplify low-concentration DNA samples [16]. Further details about this method have been elaborated in previous research [17]. PCR products validated with melt curve analysis were further confirmed using agarose gel electrophoresis.

## 3. Results

### 3.1. Study Area Description

Alipurduar was established as the 20th district in West Bengal in 2014, which has significance from the perspective of malaria transmission due to its unique location. Geographically, this district shares its borders with Assam in the east, Jalpaiguri District in the west, Cooch Behar District in the south, and the nation of Bhutan to the north. The district is characterised by a diverse landscape, with 35.5% under forest canopy and 26.8% designated as agricultural cropland. Another significant portion (15%) consists of agricultural plantations, predominantly tea estates. The region is irrigated by major rivers such as Torsa, Kaljani, Raidak, Jayanti, and Sankosh, which predominantly flow from north to south.

#### Site PHC Wise LULC Pattern

Madhya Rangalibazna PHC: The predominant land use in the adjacent villages is agricultural cropland (58%), with agricultural plantations making up 15%, while only a minor area (5.25%) is forested, and the rest are rural settlements. 

Kumargram PHC: The surrounding villages are primarily agricultural (36.23%) interspersed with village settlements (14.09%). Forested regions account for about 26.86%, while river bodies occupy around 22.23% of the total area. 

Turturi PHC: The landscape here is marked by agricultural croplands (22.90%), followed by agricultural plantations (36.94%) and village settlements (12.65%). Forest regions make up about 24.37% of the total area (Figure 1).

### 3.2. Identification of Mosquito Species Using Morphological and Molecular Methods

Morphological identification of the collected *Anopheles* species was completed by observing samples under a microscope. An image of *An. annularis s.l.*, *An. maculatus s.s.*, and *An. culicifacies s.l.* along with their respective identification characters such as wings and legs is given in Figure 2. Using allele-specific PCR, mosquito samples from the Annularis Group were identified as *An. annularis* by the characteristic PCR amplicons at 387 bp (Figure 3a). Further, for the Maculatus Group, two were determined as *An. sawadwongporni* and ten as *An. maculatus s.s.* (from now referred to as *An. maculatus*) via their PCR bands (Figure 3b). Species-specific PCR primers also helped discern three species from the Funestus Group, two as *An. aconitus*, one as *An. minimus*, and one other as *An. varuna* (Figure 3c). Where group-specific PCR was inconclusive or yielded no amplification, the (CO1) gene was amplified and sequenced for identification (Figure 3). 

From 118 mosquito samples collected over four years from the Alipurduar District, 38 were *Culex* species and two were male *Anopheles.* Details of the remaining 78 *Anopheles* specimens collected during different years of the collection are given in Table 1. In total, 7 of the 78 *Anopheles* specimens analysed were damaged, but molecular methods still identified their respective species, which were: (i) two from the Turturi PHC area in the November 2022 collection were identified as *An. sawadwongporni* (a member of the Maculatus Complex), (ii) three from the Kumargram PHC in April 2018, morphologically suspected to be *An. culicifacies s.l*., were confirmed to be the same, and (iii) of the remaining two specimens, one was molecularly identified as *An. pseudowillmori* and the other as *An. fluviatilis*. The data revealed that *An. annularis s.l.*, *An. culicifacies s.l.*, and *An. maculatus* are the major *Anopheles* species prevalent in areas of Kumargram and Turturi PHCs of Alipurduar District. In the Kumargram PHC, of the three specimens collected in 2020, in addition to *An. annularis s.l.* and *An. culicifacies s.l.*, *An. barbirostris s.l.* was also found. In the Madhya Rangalibazna PHC area, only two specimens of *An. culicifacies s.l.* were found in the samples of 2018. 

*An. annularis s.l*. was found in all three years of collection from the Kumargram PHC and the Turturi PHC. The maximum collection of 64% of the total collected of this species was from the 2022 collection from the Turturi PHC, while *An. culcifacies s.l.*, *An. maculatus*, *An. pseudowillmori*, *An. sawadwongporni*, *An. splendidus*, *An. vagus*, and *An. varuna* were all found in very small numbers in 2022 (Table 1). 

*Anopheles* species and their positivity for a human blood meal and malaria parasites are given in Table 2. Overall, the data establish that *An. annularis s.l.*, *An. culicifacies s.l.*, and *An. maculatus* are the major *Anopheles* vector species in the study areas of Alipurduar District.

### 3.3. Blood Meal Analysis

The blood meal assay amplified the desired amplicon at 519 bp, indicating the presence of human blood (Figure 3d). Most species exhibited a high positivity (50–100%) for human blood, except for *An. splendidus* (Table 2).

### 3.4. CO1 Sequencing and Phylogenetic Analysis

CO1 sequencing revealed the relationships among the vector species (Figure 4). The *An. annularis s.l.* (OR620082) DNA sequence was closest to the CO1 sequence of Thailand in the phylogenetic tree. Two *An. culicifacies s.l.* samples (OR620083 and OR620084) were sequenced, and the sample with accession no. OR620083 was found to be closest to that of Mizoram, India, and branched separately from OR620084 in the tree. The *An. maculatus* (OR620085) mosquito sample was also properly identified in the tree and was closest to that of Thailand. However, one sample that was morphologically identified as *An. maculatus* was found to be *An. pseudowillmori* (OR620087) using the CO1 sequence analysis. The *An. vagus* (OP620086) sample that was sequenced showed similarity to the CO1 sequence of Mizoram, India. Apart from this, two damaged mosquitoes, suspected to be *An. vagus*, were sequenced for further confirmation, of which one was identified as *An. fluviatilis* (OR620089), closest to that of Odisha, India, and the other as *An. pseudowillmori* (OR620088). 

### 3.5. Human Plasmodium Infection in Mosquitoes

A combination of qPCR and gel electrophoresis was used to identify *Plasmodium* infections. The infection rates varied across species, with *An. annularis s.l.* showing the highest positivity. Notably, some species had very few samples, making the infection estimates potentially less accurate for them (Table 2).

The village-wise *Anopheles* distribution data in different Sub-Centres under different PHCs of Alipurduar District are shown in Appendix A, which list the cases reported from the study area between 2018 and 2020 and in 2022.

## 4. Discussion

The Alipurduar District in West Bengal was formed in 2014, following a division of Jalpaiguri District, encompassing six blocks and spanning an area of 3136 sq. km. Both the Jalpaiguri and Alipurduar Districts have historically been hotspots for malaria, with recorded outbreaks, fatalities, and instances of drug-resistant *P. falciparum.* Factors such as the increase in vector breeding grounds, population migration into areas rich in vectors, and the introduction of new, efficient vectors have all influenced these outbreaks [6].

Historically, *An. culicifacies*, recognised as the primary vector for malaria in India, including the southern part of West Bengal, accounted for 65% of new cases, while *An. fluviatilis* was responsible for 15%. Surprisingly, neither of these species was identified as vectors in the Dooars region [18].

*An. maculatus* and *An. willmori* were previously implicated as vectors in the north-eastern states of India, such as Assam and Meghalaya, with *An. maculatus* regarded as a species with multiple forms during these times [19,20]. *An. annularis*, which was previously determined as a vector in West Bengal, India, in 1940, 1985, and 1995, has not been reported from the North Bengal region of the state since then [21]. The studies conducted previously were not performed using species-specific PCR assays, species characterisation, or molecular studies. 

Molecular studies have now pinpointed the presence of *An. pseudowillmori* and *An. sawadwongporni* from the Maculatus Group in West Bengal for the first time. Additionally, mosquito species from the Funestus Group of the Myzomyia Series, such as *An. minimus s.s*, *An. varuna*, *An. aconitus*, and *An. jeyporensis*, were confirmed with molecular methods due to their close morphological resemblances.

Emphasising the significance of combining morphological, molecular, and phylogenetic research, especially with archived samples collected by District Health Authorities, this study has unveiled the mosquito species populating an area. Our current study makes a significant contribution by identifying *An. maculatus s.s.* as a vector in West Bengal for the first time and recognising *An. culicifacies s.l.* and *An. annularis s.l.* as vectors in the Alipurduar District. As Table 1 shows, *An. annularis s.l*. emerges as the predominant species, followed by *An. maculatus s.s*. and *An. culicifacies s.l.* This prominence of *An. annularis s.l.* in the current study, notably absent in studies from 2006 to 2007, suggests its significant role in malaria transmission in the area in recent years. Historically considered a secondary vector in India’s plains, deforestation and land-use changes have possibly allowed *An. annularis* to infiltrate this sub-Himalayan region.

The presence of *An. maculatus* was documented in West Bengal’s Dooars belt based on morphology, but detailed species composition was lacking. We now confirmed the existence of *An. maculatus s.s*., *An. pseudowillmori*, and *An. sawadwongporni* from the Maculatus Group. Since we identified *An. culicifacies s.l.* and *An. annularis s.l*. as vectors, it is crucial to recognise these species at the sibling species level for a better understanding of malaria transmission. CO1 sequencing and phylogenetic analysis of two *An. culicifacies s.l.* discovered in our study indicated that they may belong to different sibling species, and further molecular analysis to differentiate between sibling species B, C, and E of the *An. culicifacies* complex is underway (unpublished data). As insecticide resistance has been observed in other region of West Bengal, it is crucial to test for this trait to optimise vector control methods in the Alipurduar region. Even with the extensive deployment of Long-Lasting Insecticide Nets (LLINs) since 2017, the continued prevalence of these vectors and related outbreaks indicates the need for ongoing research.

## 5. Conclusions

The current research on mosquitoes, collected mainly by the District Program, underscores the diverse array of vectors inhabiting this region. Our findings accentuate the need to adopt a holistic approach that merges morphological, molecular, and phylogenetic methods, particularly when working with archived specimens, to identify the mosquito species in a given region accurately and highlights the importance of collaboration between programs and research institutes.

We unveiled the presence of *An. pseudowillmori* and *An. sawadwongporni* in West Bengal, marking their debut in the state’s records. We identified *An. maculatus s.s*. as a vector in West Bengal for the first instance and identified both *An. culicifacies s.l*. and *An. annularis s.l*. as vectors in the Alipurduar region.

Considering these revelations, it becomes imperative to embark on detailed longitudinal and cross-sectional studies. A focused approach to distinguishing between sibling species and evaluating insecticide resistance becomes paramount in charting a successful path towards malaria elimination in the district.

## Figures and Tables

**Figure 1 microorganisms-12-00095-f001:**
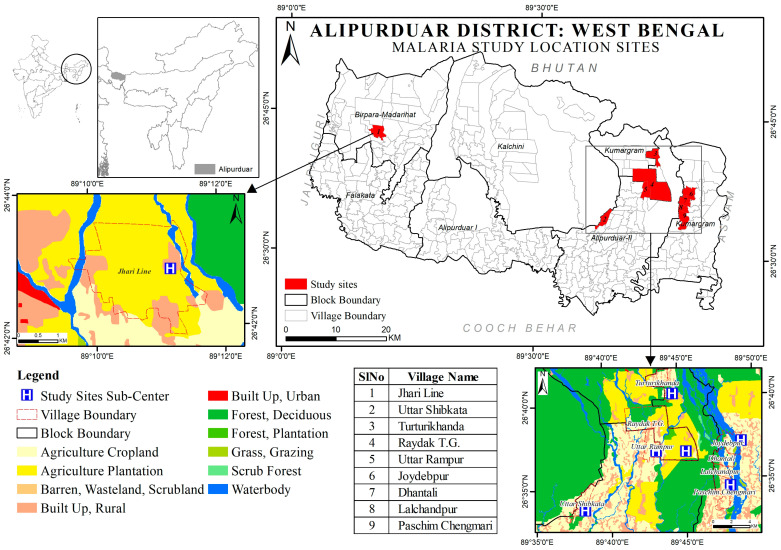
Map of Alipurduar District, West Bengal, with study areas.

**Figure 2 microorganisms-12-00095-f002:**
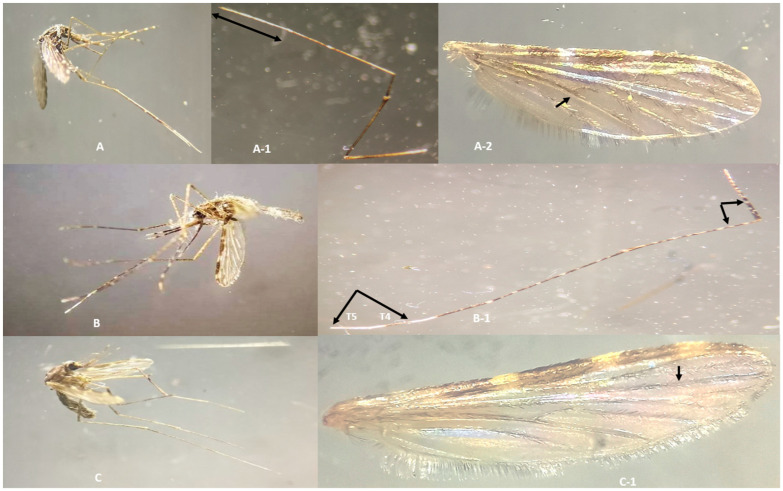
Representative images of malaria vectors of Alipurduar with important identification characters. (**A**) Adult female *Anopheles annularis s.l.*, (**A-1**) tarsomeres 3, 4, 5 of the hind leg completely white and (**A-2**) wing vein 5 dark at the point of branching. (**B**) Adult female *Anopheles (Cellia) maculatus s.l.*, (**B-1**) tarsomere 5 (T-5) and part of 4 (T-4) of the hind leg, white. Femora and tibia speckled. (**C**) Adult female *Anopheles (Cellia) culicifacies s.l.*, (**C-1**) wing vein 3, black.

**Figure 3 microorganisms-12-00095-f003:**
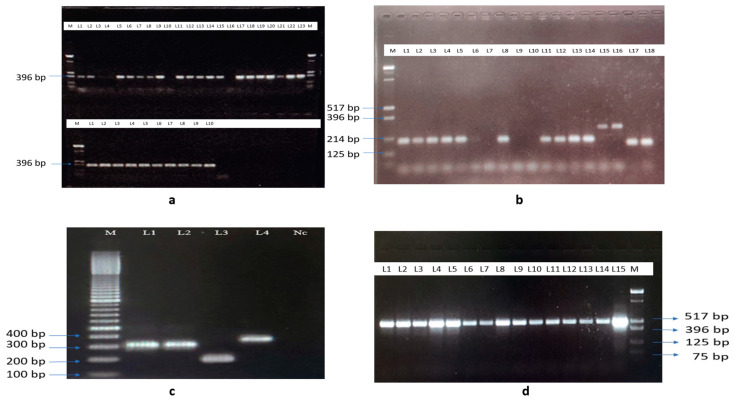
Molecular identification of different *Anopheles* species. (**a**): Characteristic PCR band of *An. annularis s.l.* at 387 bp in all the lanes except in lanes 10 and 16. M-Low molecular weight DNA ladder. Faint band in lanes 3 and 4. (**b**): Agarose gel image of PCR showing amplification of *An. maculatus s.s.* at 180 bp (lanes 1–6, 8, 11–14, and 17–18, Faint band in lane 6) and *An. sawadwongporni* at 242 bp (lanes 15 and 16). M-low molecular weight DNA ladder. (**c**): Agarose gel image of *An. aconitus* (306) bp in lane 1 and 2, lane 3: *An. minimus* (184 bp), lane 4: *An. varuna* (252 bp). M-100 bp DNA ladder, Nc-negative control. (**d**): Gel image of human blood meal PCR assay showing human blood positive at 519 bp in lanes 1 to 15. M-low molecular weight DNA ladder.

**Figure 4 microorganisms-12-00095-f004:**
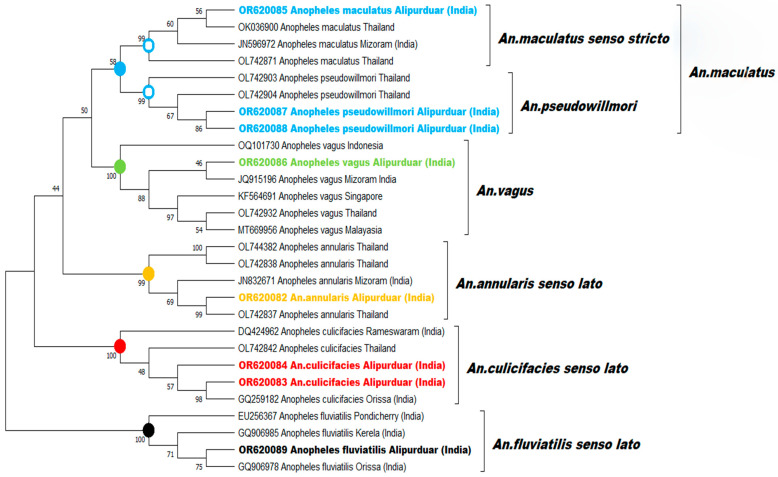
Maximum likelihood bootstrap consensus phylogenetic tree of *An. maculatus s.s*, *An. pseudowillmori*, *An. vagus*, *An. annularis s.l.*, *An. culicifacies s.l.*, and *An. fluviatilis s.l.* The tree was constructed in MEGA X by applying the general time reversible model (GTR + G + I) with 500 bootstrap values. Bold and coloured are the sequences from the current study.

**Table 1 microorganisms-12-00095-t001:** Village, Sub-Centre, and PHC-wise distribution of different mosquito species in different surveys in Alipurduar District (2018–2020, 2022).

PHC	Kumargram								Madhya Rangalibazna	Turturi				Total
Sub-Centre	Joydebpur		Turturikhanda		West Chengmari				Gopalpur T.G.	Dhowlabasti	Raydak TE-1		Uttar Shibkata	
Village	Joydebpur		Turturikhanda		Dhantali		Lalchandpur	Paschim Chengmari	Jhari Line	Uttar Rampur	Raydak TG		Uttar Shibkata	
Year (Month)	2018 (Nov)	2019 (Feb)	2018 (Apr, Nov)	2019 (Feb)	2019 (Feb)	2020 (Jul)	2019 (Feb)	2018 (Apr)	2018 (Dec)	2022 (Nov)	2018 (Oct)	2019 (Sep, Oct)	2018 (Dec)	
Total mosquito samples	1	7	8	3	7	3	3	2	2	28	2	9	3	78
*An. aconitus* (Mor, Mol)	0	0	0	0	0	0	0	0	0	0	0	2	0	2
*An. annularis s.l.* (Mor, Mol)	1	2	2	0	3	1	0	0	0	18	0	6	1	34
*An. barbirostris s.l.* (Mor)	0	0	0	1	0	1	0	0	0	0	0	0	1	3
*An. culicifacies s.l.* (Mor, Mol)	0	0	5	0	0	1	2	1	2	2	0	0	0	13
*An. fluviatilis* (Mol)	0	0	1	0	0	0	0	0	0	0	0	0	0	1
*An. hyrcanus s.l.* (Mor)	0	0	0	0	0	0	0	0	0	0	1	0	0	1
*An. maculatus s.s.* (Mor, Mol)	0	4	0	2	4	0	1	0	0	2	0	0	0	13
*An. minimus* (Mor, Mol)	0	0	0	0	0	0	0	0	0	0	0	1	0	1
*An. pseudowillmori* (Mol)	0	1	0	0	0	0	0	0	0	1	0	0	0	2
*An. sawadwongporni* (Mol)	0	0	0	0	0	0	0	0	0	2	0	0	0	2
*An. splendidus* (Mor)	0	0	0	0	0	0	0	0	0	1	0	0	0	1
*An. vagus* (Mor)	0	0	0	0	0	0	0	1	0	1	1	0	1	4
*An. varuna* (Mor, Mol)	0	0	0	0	0	0	0	0	0	1	0	0	0	1

**Table 2 microorganisms-12-00095-t002:** Malaria parasite (Pf—*Plasmodium falciparum*, Pv—*Plasmodium vivax*) positivity and human blood meal positivity of mosquito species identified in the Alipurduar District of West Bengal (2018–2020, 2022).

Mosquito Species	Total Samples Tested with PCR	Pf (%)	Pv (%)	Mixed (Pf + Pv) (%)	Total Plasmodium-Positive (%)	Human Blood Meal Positivity Using Exclusive PCR for Human Blood
*An. aconitus*	2	0 (0)	1 (50)	1 (50)	2 (100)	1.0
*An. annularis s.l.*	34	7 (20.59)	5 (14.71)	9 (26.47)	21 (61.76)	0.76
*An. barbirostris s.l.*	3	0 (0)	1 (33.33)	0 (0)	1 (33.33)	1.0
*An. culicifacies s.l.*	13	1 (7.69)	1(7.69)	0 (0)	2 (15.38)	0.92
*An. fluviatilis*	1	0 (0)	0 (0)	0 (0)	0 (0)	1.0
*An. hyrcanus s.l.*	1	1 (100)	0 (0)	0 (0)	1 (100)	1.0
*An. maculatus s.s.*	13	2 (15.38)	2 (15.38)	2 (15.38)	6 (46.15)	0.92
*An. minimus*	1	0 (0)	1 (100)	0 (0)	1 (100)	1.0
*An. pseudowillmori*	2	0 (0)	0 (0)	0 (0)	0 (0)	0.5
*An. sawadwongporni*	2	0 (0)	0 (0)	0 (0)	0 (0)	0.5
*An. splendidus*	1	0 (0)	0 (0)	0 (0)	0 (0)	0
*An. vagus*	4	1 (25)	0 (0)	0 (0)	1 (25)	0.5
*An. varuna*	1	0 (0)	0 (0)	0 (0)	0 (0)	1.0
Total	78	12 (15.38)	11(14.10)	12 (15.38)	35 (44.87)	0.81

## Data Availability

The data presented in this study are available upon request from the corresponding author. The data are not publicly available due to ethical and privacy reasons.

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
