# Peer review of "First Report of Anopheles annularis s.l., An. maculatus s.s., and An. culicifacies s.l. as Malaria Vectors and a New Occurrence Record for An. pseudowillmori and An. sawadwongporni in Alipurduar District Villages, West Bengal, India"

_microorganisms, 2024, doi:10.3390/microorganisms12010095_

Round 1
Reviewer 1 Report
Comments and Suggestions for Authors
The manuscript I received for review is clear, written very well substantively and formally, and in correct English. In my opinion, the manuscript is worth publishing.
The only thing that needs improvement in my opinion is a large number of minor editorial flaws, which I will list below:
L2 – no space between maculatus and s.s.
I highly recommend thoroughly checking the entire manuscript for similar flaws (e.g. Line 41, L44, L51, L62,L161, L205, L207, L210 and many, many more)
L3 – it looks like double space between vectors and and
L57 – why are these green colors in keywords?
L57 – should it say "Malaria Elimination" in capital letters?
L72 – for the first and second References a footnote was used in [], but later () throughout the manuscript - change it please
Figure 1 – I strongly recommended to label the map better, these two parts without color
L153 – ribosomal
Figure 3 – If possible, make the Figure more clear and sharp, the print is blurry and very difficult to read
L227 – double space (the same L255)
L228 - fluviatilis
Table 2 – explain abbreviations Pf and Pv
The discussion section is missing line numbers so I can't comment on a specific line, italics are missing in An. maculatus
Fig 2b – lane or Lane – be consequent
I think the manuscript is very good and further research is promising.
Author Response
Respected Reviewer, first of all, we would like to thank you very much for taking a keen interest in reviewing the manuscript. We appreciate your comment on the manuscript; it has helped immensely to rectify all the editorial flaws that were present earlier. We are providing the point-to-point response to your comments in the word document attached.

Reviewer 2 Report
Comments and Suggestions for Authors
Dear authors, I've read the manuscript "First report of Anopheles annularis s. l., An. maculatuss.s. and An. culicifacies s.l. as malaria vectors and new occurrence record of An.pseudowillmori and An.swadangporni in villages of District Alipurduar, West Bengal, India”. My opinion is that is very interesting and valuable and it has to be published. Unfortunately my suggestion is to reject the manuscript in present form. You should correct it and if you want, re-submit it for publication. Here are some of my concerns that should be corrected:
1) Manuscript should be checked by native speaker or English lector as some sentences are not understandable for the reader.
2) Title should be changed into “First report of Anopheles annularis s. l., An. maculatus s. s. and An. culicifacies s. l. as malaria vectors and new occurrence record for An. pseudowillmori and An. swadangporni in Alipurduar district villages, West Bengal, India”
3) all the other suggestions that had to be done are in the “corrected” version of the manuscript
Best regards

Author Response
Respected reviewer, thanks for the suggestion that has been given. Considering your comments, we have taken the help of some good English speaker and all the necessary changes and improvement in the manuscript has been made. Your suggestions on the manuscript were useful in reshaping and rewriting some of the paragraphs and the editorial flaws. We are sending the point-to-point response to your comments in the word document attached.

Reviewer 3 Report
Comments and Suggestions for Authors
Dear Authors,
Although your findings are very interesting, the quality of the written material is very poor. The language of the Manuscript requires significant improvement.
The technical side of MS is not good. Please revise all again. For example: there is missing space after the full stop in lines 41, 44, 45, 52, etc. I found it in a lot of places in the text.
The citation in brackets is not formatted. I suppose that number 3 in line 82 and number 4 in line 91 are citations. If so, they should be in square brackets.
Keywords are not adequate such as “first occurrence of”. There is some color which is left in the keywords.
All species given in Latin names should be in italics.
Lines 51-53 should be completely removed because it is something that does not belong there. To general.
The first paragraph of the introduction belongs to the Material and Methods.
L104 What do you mean by “conspicuously” missing? There are two possibilities: missing and not missing.
Material and Method: It is not well described how the sampling was carried out. What do you mean by hand? What method is that? The authors gave more than needed details about the area but nothing about the sampling.
L151 This is not the correct citation format. Please check the journal’s propositions.
L151 Phuc et al what? Protocol? Method? Key?
Figure 3. Some lines are very blurry.
Lines 181-188 This should be given in the Material and Methods
In the Results section authors should first give how many mosquitoes they collected, how many were analyzed, species distribution in different regions or villages, etc. It should be significantly improved.
L232 In several cases there is a capital letter of the name of species because of the automatic capital letter after a full stop such as An. Annularis.
Regardless the authors obtained interesting results, there is no structure in the MS. Therefore, it is necessary to revise it before submitting it again.
Comments on the Quality of English LanguageThe language must be improved.
Author Response
Respected Reviewer, we are very thankful for the suggestion and important comments on the manuscript. Taking account of your suggestion we have edited the manuscript and all the editorial and technical flaws were rectified. We are providing the point-to-point response to your comment in the word document attached.

Reviewer 4 Report
Comments and Suggestions for Authors
1. material and methods should be rewritten to be reproducible. 2. The conclusion needs to be short and to the point (rewrite again).
3. The manuscript needs linguistic editing.
Comments on the Quality of English LanguageThe author considered all revisions than the original one.
Author Response
We are thankful to the editors and reviewers for taking keen interest in this work. We have attempted to present the methods and results in lucid manner and discuss the study considering the obtained results. There were some comments raised by the reviewer 4, which we have addressed and the manuscript has been revised accordingly. The point-wise reply to the comments are as follows:
Comment 1: material and methods should be rewritten to be reproducible.
Response 1: This section of the manuscript has been restructured. We have added the methodology of map preparation from L112 to L119. Mosquito collection method is added in L121-l24. We have also added the PCR primers details in supplementary table S3.
Comment 2: The conclusion needs to be short and to the point (rewrite again).
Response 2: The conclusion part has been rewritten as suggested and presented in L346-L359 in the revised manuscript.
Comment 3: The manuscript needs linguistic editing.
Response 3: We took suggestion of some good English speaker and taking consideration of all the reviewers we have restructured some paragraphs and sentences of the manuscript.
Comment 4: The author considered all revisions than the original one.
Response 4: In line with the suggestions, the manuscript has been revised
Round 2
Reviewer 2 Report
Comments and Suggestions for Authors
Dear Authors, thank you for improving your manuscript :-)
Best regards
Reviewer 3 Report
Comments and Suggestions for Authors
Dear Authors,
After you accepted the suggestions your MS is significantly improve. I recommed it to be published
Comments on the Quality of English LanguageEnglish is fine. No major corrections required.
Reviewer 4 Report
Comments and Suggestions for Authors
The authors addressed the comment
Comments on the Quality of English LanguageThe English quality improved.